# From tests to truth: A misclassification-aware machine learning framework for estimating brucellosis seroprevalence in wild canids

Nahal Sarvestani[1☉], Farzane Shams[2☉], Armin Mirshahi[3], Mobina Pato[1],
Aria Javani Farbod[4], Armina Khayatderafshi[5], Mobina Payami[6], Arman Abdous [6,7]*

1 Faculty of Veterinary Medicine, Science and Research Branch, Islamic Azad University, Tehran, Iran,
2 Division of Neurological Sciences, Vetsuisse Faculty, University of Bern, Bern, Switzerland, 3 Faculty
of Veterinary Medicine, Garmsar Branch, Islamic Azad University, Semnan, Iran, 4 Faculty of Veterinary
Medicine, Urmia Branch, Islamic Azad University, Urmia, Iran, 5 Veterinary Practice Manager, North Care
Animal Hospital, North Vancouver, British Columbia, Canada, 6 Faculty of Veterinary Medicine, Karaj
Branch, Islamic Azad University, Karaj, Iran, 7 School of Public Health, Tehran University of Medical
Sciences, Tehran, Iran

☉ These authors contributed equally to this work.
* Armanabds1374@gmail.com

## Abstract

Brucellosis is an important zoonotic disease affecting humans, livestock, and wild-life, yet prevalence estimates in wild species are often underestimated due to limited attention to surveillance, as well as insufficient and biased sampling. To clarify exposure patterns in wild canids, we searched PubMed, Scopus, Web of Science, SciELO, and Google Scholar (1962–2025) for primary prevalence studies of Brucella species (spp.) in free-ranging canids. Serological data were analyzed using a misclassification-aware, multi-assay model that corrects for imperfect test sensitivity and specificity. Confirmatory polymerase chain reaction (PCR) and culture results were considered separately from serological data, as each offers a different perspective on disease status. Across 48 wild serology populations (n = 3,925 animals), the global misclassification-adjusted true seroprevalence was 8.2% (95% confidence interval [CI]: 5.1–11.3%). Confirmed active infection, based on PCR or culture, was uncommon (3.9%; 95% CI: 3.0–5.1%). Exposure levels varied across continents, with higher estimates in South America (approximately 18%) and lower levels in Europe (approximately 0.8%) and North America (approximately 4.1%). Data from Africa were limited, and Asian estimates were based on sparse wild samples, leading to wide uncertainty. Seroprevalence was consistently influenced by assay type, host species, and region. Overall, wild canids exhibit modest but widespread serological exposure to Brucella spp., whereas confirmed active infection remains rare. Because evidence quality and diagnostic rigor varied considerably across regions, especially in Africa and parts of Asia, results from these areas should be interpreted with caution. By correcting for imperfect tests and explicitly accounting for study

**Data availability statement:** All analyses were performed using Python. The complete analysis scripts and associated datasets are publicly available at the ShamsFarzane GitHub repository: https://github.com/ShamsFarzane/brucellosis-canids-seroprevalence.git.

**Funding:** The author(s) received no specific funding for this work.

**Competing interests:** The authors have declared that no competing interests exist.

heterogeneity, this framework provides more reliable and transparent prevalence estimates and highlights geographic gaps where improved, targeted One Health surveillance is most urgently needed.

## Author summary

Brucellosis is a bacterial disease that affects livestock, wildlife, and people. While surveillance usually focuses on domestic animals, far less is known about infection in wild carnivores such as wolves, foxes, and jackals, even though these species often live near farms and may signal when *Brucella* is present in the environment. Research on wild canids is difficult to compare because studies use many different diagnostic tests, some of which can give false-positive or false-negative results. We reviewed all available studies of *Brucella* in wild canids worldwide and combined the data using a modeling approach that adjusts for imperfect diagnostic accuracy. We found that exposure is widespread but unevenly documented, with major gaps in Africa and much of Asia. After accounting for test performance, wild canids showed modest levels of exposure overall, and confirmed infections were rare. Studies using older or less reliable tests tended to report inflated prevalence. By correcting for diagnostic limitations, our study provides more reliable estimates and highlights where improved wildlife surveillance is most needed to support One Health monitoring.

## Introduction

Brucellosis is a globally distributed zoonotic disease of major veterinary and public health concern, caused by Gram-negative bacteria of the genus *Brucella* [1]. These pathogens infect a wide range of mammals, including livestock, humans, and wildlife. While most control and surveillance programs have primarily focused on livestock and human infections, wildlife, especially free-ranging canids such as wolves (*Canis lupus*), foxes (*Vulpes spp.*, *Lycalopex spp.*), jackals (*Canis aureus*), and coyotes (*Canis latrans*), may also contribute to the ecology and transmission of *Brucella* species [2]. As predators and scavengers, wild canids may act as incidental hosts, spillover recipients, or mechanical vectors, and in certain contexts, could act as local reservoirs. These dynamics are most relevant in ecotones and rural or peri-urban landscapes, where land-use change increases contact among wildlife, livestock, and humans.

Despite the zoonotic importance of *Brucella*, surveillance in wild canids remains uneven. Most published studies originate from the Americas, while large data gaps persist in Africa, Asia, and Eastern Europe [3]. Much of the existing field research relies on serological tests such as the Standard Agglutination Test (SAT), Rose Bengal Test (RBT), and enzyme-linked immunosorbent assay (ELISA) because they are affordable and practical, yet these methods primarily reflect past exposure rather than active infection [4]. In contrast, bacterial culture and polymerase chain reaction (PCR)

provide stronger evidence of active infection but are costly and logistically challenging, which has limited their routine use in wildlife studies [5]. Distinguishing clearly between seroprevalence (exposure) and PCR/culture-confirmed infection is therefore essential for understanding transmission potential and designing targeted surveillance systems.

Previous reviews of wildlife brucellosis by Dadar *et al.* and Kosoy and Goodrich have synthesized data across species and diagnostic methods, but they often lacked species-specific analyses, formal quality assessment, and statistical correction for imperfect diagnostic performance [3,6]. These limitations, combined with sparse and geographically uneven datasets, make it difficult to draw reliable conclusions about exposure patterns in wild canids.

To address these gaps, we focused specifically on the family Canidae (globally distributed, mobile species that commonly interact with livestock) and applied a reproducible, quantitative framework that builds on established misclassification and latent-class approaches. We extended these approaches to handle challenges such as multi-assay testing within the same sampled populations, heterogeneous study designs, and variable diagnostic quality. Using a PRISMA-ScR-guided scoping review and a structured quality assessment adapted from the Newcastle–Ottawa Scale, we first characterized the global evidence base and identified taxonomic and geographic gaps. Second, we implemented a misclassification-aware model that integrates results from multiple serological assays to estimate true population-level seroprevalence, rather than treating each assay arm independently, and we analyzed PCR/culture data separately. Third, we applied an exploratory machine learning meta-regression to identify consistent predictors of seroprevalence such as assay type, species, and region to support practical surveillance design tasks, including sample size estimation under imperfect testing conditions. Together, these components synthesize existing methodological principles into an integrated, wildlife-focused framework that produces uncertainty-aware, species- and region-specific prevalence estimates to inform One Health surveillance and control strategies at the wildlife–livestock–human interface.

## Methods

### Bibliographic search strategy

We conducted a global scoping review and quantitative synthesis of *Brucella* exposure and infection in wild canids (family *Canidae*), following the PRISMA Extension for Scoping Reviews (PRISMA-ScR) guidelines [7]. The study protocol defined eligibility, screening, data extraction, and analysis steps *a priori* and was piloted before full implementation. A PRISMA flow diagram summarizes record identification, screening, and full-text assessment (Fig 1), with detailed reasons for full-text exclusion provided in S1 Table. The completed PRISMA-ScR checklist is provided in S1 File.

A comprehensive literature search was conducted in July 2025 across PubMed, Scopus, Web of Science Core Collection, SciELO, and Google Scholar. The search strategy combined controlled vocabulary, where available, and free-text keywords related to *Brucella* infection, wild canids, epidemiology, and methods. Based on pilot testing, the first 200 Google Scholar records were screened for relevance. Detailed search strings, filters, and database-specific syntax are provided in S2 File. As a representative example, the PubMed search strategy was: ("Brucellosis"[Mesh] OR "Brucella abortus"[Mesh] OR "Brucella suis"[Mesh] OR "Brucella canis"[Mesh] OR brucellosis OR Brucella) AND ("Canidae"[Mesh] OR canids OR "wild canids" OR foxes OR wolves OR jackals OR coyotes OR "wild dogs") AND (prevalence OR seroprevalence OR serology OR ELISA OR PCR OR culture OR diagnostics). All retrieved records were de-duplicated in a unified library before screening.

Two reviewers independently screened titles and abstracts, followed by full-text assessment using prespecified eligibility criteria. Disagreements were resolved by consensus, and inter-rater agreement was assessed during a calibration exercise on a random subset of studies using Cohen's kappa (κ).

### Data extraction and synthesis

A piloted extraction form was used to record metadata from each included study, including author, sampling year(s), country and continent, host species (common and scientific names), diagnostic methods, life condition (wild or captive), sample

 

PLOS Neglected Tropical Diseases

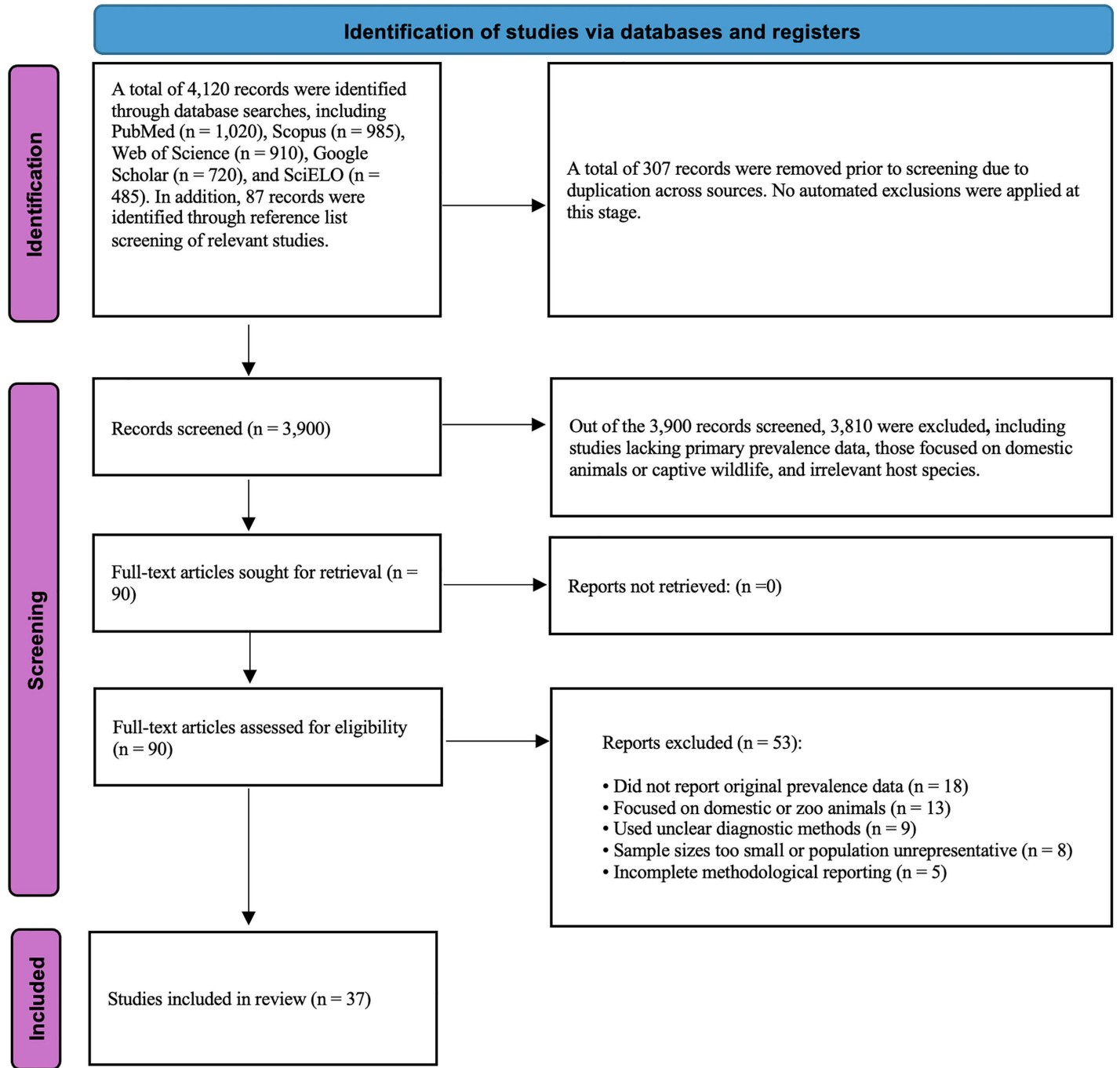

**Fig 1. PRISMA-ScR flow diagram of study selection for the global scoping review of *Brucella* exposure and infection in wild canids (1962–2025).**

size (N), number of positives (n), and the *Brucella* species reported, when available. Species names were standardized according to current *Canidae* taxonomy, and country names were harmonized using International Organization for Standardization (ISO) formats to ensure consistency across studies.

Each study was assigned a unique Study ID composed of the first author's surname and publication year, for example Davis_1979. Records sharing the same Study ID and sampling year(s) were treated as the same study, and duplicate records retrieved from multiple databases were merged. When a single publication reported results for multiple species, diagnostic methods, or sampling periods, each unique combination was treated as a distinct study arm.

Study arms were collapsed only when they clearly referred to the same animals, indicated by identical species, country, sampling period, and total sample size (N). This occurred primarily when multiple serological assays such as RBT, SAT, or ELISA were applied to the same individuals. In such cases, the number of positives and total sample size were summed and treated as a multi-test population. Study arms differing in species, country, sampling year(s), or N, or where overlap of individuals was uncertain, were retained as separate populations.

A strict separation was maintained between diagnostic tracks. Serological assays were used to assess exposure, whereas PCR and bacterial culture were used to identify active infection. These diagnostics were applied to different individuals in nearly all included studies and were therefore never combined. Each population was assigned exclusively to either the serology track or the PCR and culture track. Study-level prevalence data derived from this process are summarized in Table 1.

## Inclusion and exclusion criteria

Records were eligible if they reported primary prevalence data for wild-caught canids, including wolves (*Canis lupus*), foxes (*Vulpes spp.*, *Lycalopex spp.*), jackals (*Canis aureus*), coyotes (*Canis latrans*), African wild dogs (*Lycaon pictus*), and used validated diagnostic methods. No restrictions on publication year or language were applied during database searches, and the final body of eligible evidence spanned publications from 1962 to 2025. All retrieved records were screened regardless of date, language, or region of origin.

Studies were eligible if they reported primary prevalence data for wild-caught canids and provided sufficient methodological detail to confirm host identity and diagnostic procedures.Diagnostics were categorized a priori into two groups:

1. **Serology (exposure):** Standard Agglutination Test (SAT), Rose Bengal Test (RBT), Complement Fixation Test (CFT), agar gel immunodiffusion (AGID), enzyme-linked immunosorbent assay (ELISA), Fluorescence Polarization Assay (FPA), or equivalent antibody-based assays.

2. **Active infection (confirmation):** Polymerase chain reaction (PCR), including quantitative PCR, and/or bacterial culture.

Studies involving zoo or laboratory animals were excluded, as were reports lacking primary numerators and denominators or sufficient information to verify host species identity or diagnostic methodology.

## Quality appraisal (QA)

Study quality was evaluated using an adapted Newcastle–Ottawa Scale for prevalence studies [45].

Because the original NOS was developed for cohort and case–control designs, the scale was adapted to address sources of bias relevant to wildlife serosurveys and molecular diagnostic studies. The adapted NOS included seven binary items grouped into three domains: selection (S1–S3), comparability/diagnostic quality (C1–C2), and outcome reporting (O1–O2).

Each item was scored as 1 (criterion met) or 0 (criterion not met), giving a total score ranging from 0 to 7, consistent with standard NOS-based scoring approaches for prevalence studies. The summed score for each study is reported in the final "QA" column, where higher values indicate lower risk of bias. Based on total scores, studies were categorized as having low (5–7 points), moderate (3–4 points), or high (0–2 points) risk of bias, reflecting increasing concern about internal validity.

**Table 1. Study-level brucellosis prevalence in wild canids worldwide (1962–2025).**

| | Study ID | Year of Sampling | Country | Continent | Diagnostic Category | Life Condition | Diagnostic Method | Type of Brucellosis | Positive | N (sample size) | Species | QA |
|---|---|---|---|---|---|---|---|---|---|---|---|---|
| [8] | Proença, L. | 2006 | Brazil | South America | Serology | Wild | RBT | Brucella abortus | 0 | 3 | Maned wolf (Chrysocyon brachyurus) | 2 |
| [8] | Proença, L. | 2006 | Brazil | South America | Serology | Wild | RBT | Brucella abortus | 0 | 7 | Crab-eating fox (Cerdocyon thous) | 2 |
| [9] | Martino, P. E | 1998-2001 | Argentina | South America | Serology | Wild | ELISA | Brucella spp | 8 | 28 | Culpeo fox (Dusicyon culpaeus) | 7 |
| [9] | Martino, P. E | 1998-2001 | Argentina | South America | Serology | Wild | ELISA | Brucella spp | 7 | 56 | Culpeo fox (Dusicyon culpaeus) | 7 |
| [10] | Oliveira-Filho, E. F | 2006 | Brazil | South America | Serology | Wild | CFT | Brucella spp | 8 | 42 | Fox spp. | 6 |
| [10] | Oliveira-Filho, E. F | 2006 | Brazil | South America | Serology | Wild | RBT | Brucella spp | 23 | 42 | Fox spp. | 6 |
| [10] | Oliveira-Filho, E. F | 2006 | Brazil | South America | Serology | Wild | AGID | Brucella spp | 7 | 42 | Fox spp. | 6 |
| [11] | Dorneles, E. M. S. | 2005-2009 | Brazil | South America | Serology | Wild | RBT | Brucella spp | 5 | 38 | Crab-eating fox (Cerdocyon thous) | 6 |
| [12] | Uzai, G. J. S. | 2018 | Brazil | South America | Molecular | Wild | PCR | Brucella spp | 0 | 20 | Crab-eating fox (Cerdocyon thous) | 6 |
| [13] | de Azevedo, S. S. | 2004 | Brazil | South America | Serology | Wild | BPAT | Brucella abortus | 16 | 60 | Pseudalopex vetulus (hoary fox) | 7 |
| [13] | de Azevedo, S. S. | 2004 | Brazil | South America | Serology | Wild | 2-ME | Brucella abortus | 4 | 60 | Pseudalopex vetulus (hoary fox) | 7 |
| [14] | Galarce, N. | 2020 | Chile | South America | Serology | Wild | CIEF | Brucella canis | 5 | 46 | Fox spp. | 6 |
| [15] | Galarce Gálvez, N. E | 2020 | Chile | South America | Serology | Wild | CIEF | Brucella canis | 5 | 31 | Culpeo fox (Lycalopex culpaeus) | 6 |
| [15] | Galarce Gálvez, N. E | 2020 | Chile | South America | Serology | Wild | CIEF | Brucella canis | 0 | 14 | Chilla fox (Lycalopex griseus) | 6 |
| [15] | Galarce Gálvez, N. E | 2020 | Chile | South America | Serology | Wild | CIEF | Brucella canis | 0 | 4 | Red fox (Vulpes vulpes) | 6 |
| [15] | Galarce Gálvez, N. E | 2020 | Chile | South America | Serology | Wild | CIEF | Brucella canis | 0 | 4 | Wolf (Canis lupus) | 6 |
| [16] | Moya, S. | 2008-2012 | Chile | South America | Serology | Wild | CIEF | Brucella canis | 0 | 15 | Fuegian culpeo fox (Pseudalopex culpaeus lycoides) | 6 |
| [16] | Moya, S. | 2008-2012 | Chile | South America | Serology | Wild | CIEF | Brucella canis | 0 | 12 | Chilla fox (Pseudalopex griseus) | 6 |
| [11] | Dorneles, E. M. S. | 2005-2009 | Brazil | South America | Serology | Wild | FPA | Brucella spp | 5 | 38 | Crab-eating fox (Cerdocyon thous) | 6 |
| [17] | Hidalgo-Hermoso, E. | 2013-2018 | Chile | South America | Serology | Wild | ELISA | Brucella abortus | 0 | 46 | Darwin's fox (Lycalopex fulvipes) | 6 |
| [17] | Hidalgo-Hermoso, E. | 2013-2018 | Chile | South America | Serology | Wild | CIE | Brucella canis | 0 | 46 | Darwin's fox (Lycalopex fulvipes) | 6 |
| [18] | Fiorello, C. V. | 2001-2005 | Bolivia | South America | Serology | Wild | AGID | Brucella canis | 0 | 9 | Pampas fox (Pseudalopex gymnocercus) | 4 |
| [18] | Fiorello, C. V. | 2001-2005 | Bolivia | South America | Serology | Wild | AGID | Brucella canis | 0 | 5 | Crab-eating fox (Cerdocyon thous) | 4 |
| [19] | Szyfres, B. | 1962–1964 | Argentina | South America | Serology | Wild | SAT | Brucella abortus | 173 | 728 | Fox spp. | 3 |
| [20] | de Macedo, G. C. | 2021 | Brazil | South America | Serology | Wild | RBT | Brucella spp | 5 | 38 | Crab-eating fox (Cerdocyon thous) | 4 |

*(Continued)*

| | Study ID | Year of Sampling | Country | Conti-nent | Diagnostic Category | Life Condition | Diagnos-tic Method | Type of Brucellosis | Pos-itive | N (sam-ple size) | Species | QA |
|---|---|---|---|---|---|---|---|---|---|---|---|---|
| [21] | Chitwood, M. C. | 2011 | USA | North America | Serology | Wild | IFA | Brucella canis | 0 | 30 | Coyote (Canis latrans) | 6 |
| [21] | Chitwood, M. C. | 2011 | USA | North America | Serology | Wild | RIV | Brucella abortus | 0 | 28 | Coyote (Canis latrans) | 6 |
| [22] | Williams, J. D. | 1988 | USA | North America | Serology | Wild | SAT | Brucella abortus | 17 | 94 | Coyote (Canis latrans) | 7 |
| [22] | Williams, J. D. | 1988 | USA | North America | Serology | Wild | CARD | Brucella abortus | 38 | 94 | Coyote (Canis latrans) | 7 |
| [22] | Williams, J. D. | 1988 | USA | North America | Serology | Wild | CFT | Brucella abortus | 21 | 94 | Coyote (Canis latrans) | 7 |
| [22] | Williams, J. D. | 1988 | USA | North America | Serology | Wild | ELISA | Brucella abortus | 29 | 94 | Coyote (Canis latrans) | 7 |
| [22] | Williams, J. D. | 1988 | USA | North America | Serology | Wild | RIV | Brucella abortus | 20 | 94 | Coyote (Canis latrans) | 7 |
| [23] | Zarnke.R | 1975-1998 | USA | North America | Serology | Wild | BBA | Brucella suis type 4 | 27 | 930 | Wolf (Canis lupus) | 3 |
| [23] | Zarnke.R | 1975-1982 | USA | North America | Serology | Wild | BBA | Brucella suis type 4 | 1 | 67 | Wolf (Canis lupus) | 3 |
| [24] | Hoq, M. A | 1977 | USA | North America | Serology | Wild | RBT | Brucella abortus | 9 | 148 | Coyote (Canis latrans) | 3 |
| [25] | Hoff, G. L. | 1973 | USA | North America | Serology | Wild | SAT | Brucella canis | 1 | 68 | Red fox (Vulpes vulpes) | 2 |
| [25] | Hoff, G. L. | 1973 | USA | North America | Serology | Wild | SAT | Brucella canis | 0 | 15 | Gray fox (Urocyon cinereoargenteus) | 2 |
| [25] | Hoff, G. L. | 1973 | USA | North America | Serology | Wild | SAT | Brucella canis | 2 | 103 | Coyote (Canis latrans) | 2 |
| [25] | Hoff, G. L. | 1973 | USA | North America | Serology | Wild | SAT | Brucella canis | 0 | 4 | Wolf (Canis lupus) | 2 |
| [26] | Randhawa, A. S. | 1976 | USA | North America | Serology | Wild | 2-ME | Brucella canis | 16 | 198 | Coyote (Canis latrans) | 4 |
| [27] | Davis, D. S. | 1978 | USA | North America | Serology | Wild | SAT | Brucella abortus | 9 | 51 | Coyote (Canis latrans) | 3 |
| [28] | Neiland, K. A. | 1969 | USA | North America | Serology | Wild | CFT | Brucella suis type 4 | 3 | 7 | Wolf (Canis lupus) | 3 |
| [29] | Tessaro, S. V. | 1986 | Canada | North America | Serology | Wild | RBT | Brucella abortus | 1 | 37 | Red fox (Vulpes vulpes) | 2 |
| [29] | Tessaro, S. V. | 1986 | Canada | North America | Serology | Wild | RBT | Brucella abortus | 4 | 13 | Wolf (Canis lupus) | 2 |
| [30] | Neiland, K. A. | 1967–1972 | USA | North America | Serology | Wild | CFT | Brucella suis type 4 | 11 | 28 | Wolf (Canis lupus) | 4 |
| [30] | Neiland, K. A. | 1967–1972 | USA | North America | Serology | Wild | CFT | Brucella suis type 4 | 2 | 11 | Red fox (Vulpes vulpes) | 4 |
| [31] | Schnurren-berger, P. R | 1977–1983 | USA | North America | Serology | Wild | SAT | Brucella abortus | 0 | 2 | Coyote (Canis latrans) | 1 |
| [31] | Schnurren-berger, P. R | 1977–1983 | USA | North America | Serology | Wild | SAT | Brucella abortus | 1 | 47 | Gray fox (Urocyon cinereoargenteus) | 1 |
| [32] | Morton, J. K | 1988 | Alaska | North America | Serology | Wild | SAT | Brucella suis type 4 | 12 | 64 | Arctic fox (Vulpes lagopus) | 3 |
| [33] | McCue, P. M. | 1981-1984 | USA | North America | Serology | Wild | CFT | Brucella abortus | 4 | 65 | San Joaquin kit fox (Vulpes macrotis mutica) | 6 |

*(Continued)*

**Table 1.** (Continued)

| | Study ID | Year of Sampling | Country | Conti-nent | Diagnostic Category | Life Condition | Diagnos-tic Method | Type of Brucellosis | Pos-itive | N (sam-ple size) | Species | QA |
|---|---|---|---|---|---|---|---|---|---|---|---|---|
| [33] | McCue, P. M. | 1981-1984 | USA | North America | Serology | Wild | CFT | Brucella canis | 5 | 56 | San Joaquin kit fox (Vulpes macrotis mutica) | 6 |
| [34] | Ćirović, D. | 2010-2013 | Serbia | Europe | Molecular | Wild | PCR | Brucella canis | 4 | 216 | Golden jackal (Canis aureus) | 7 |
| [35] | I.V. Nakonechnyi | 2018 | Ukraine | Europe | Serology | Wild | RBT | Brucella spp | 0 | 9 | Golden jackal (Canis aureus) | 5 |
| [36] | Bertelloni, F. | 2023 | Italy | Europe | Molecular | Wild | PCR | Brucella suis type 4 | 0 | 16 | Wolf (Canis lupus) | 6 |
| [37] | Nymo, I. H. | 1995–2003 | Norway | Europe | Serology | Wild | ELISA | Brucella spp | 0 | 406 | Arctic fox (Vulpes lagopus) | 7 |
| [38] | Ebani, V. V. | 2016-2021 | Italy | Europe | Molecular | Wild | PCR | Brucella spp | 0 | 22 | Red fox (Vulpes vulpes) | 6 |
| [39] | Minichino et al. | 2023–2024 | Italy | Europe | Serology | Wild | ICT | Brucella canis | 5 | 18 | Red fox (Vulpes vulpes) | 6 |
| [39] | Minichino et al. | 2023–2024 | Italy | Europe | Serology | Wild | RBT | Brucella spp | 2 | 18 | Red fox (Vulpes vulpes) | 6 |
| [39] | Minichino et al. | 2023–2024 | Italy | Europe | Serology | Wild | ICT | Brucella canis | 0 | 4 | Wolf (Canis lupus) | 6 |
| [39] | Minichino et al. | 2023–2024 | Italy | Europe | Serology | Wild | RBT | Brucella spp | 0 | 4 | Wolf (Canis lupus) | 6 |
| [40] | Zhou, Y. | 2017 | China | Asia | Serology | Farm | RBT | Brucella melitensis | 493 | 726 | Blue fox (Vulpes lagopus) | 5 |
| [40] | Zhou, Y. | 2017 | China | Asia | Serology | Farm | SAT | Brucella melitensis | 300 | 726 | Blue fox (Vulpes lagopus) | 5 |
| [41] | Egorov, E. A. | 1996 | Russia | Asia | Culture | Wild | Culture | Brucella suis type 4 | 30 | 254 | Wolf (Canis lupus) | 4 |
| [41] | Egorov, E. A. | 1996 | Russia | Asia | Culture | Wild | Culture | Brucella suis type 4 | 18 | 777 | Arctic fox (Vulpes lagopus) | 4 |
| [42] | Lapid, R. | 2022 | Israel | Asia | Serology | Wild | RBT | Brucella spp | 0 | 56 | Golden jackal (Canis aureus) | 5 |
| [43] | Pinigin, A. F. | 1969 | Russia | Asia | Serology | Farm | SAT | Brucella suis type 4 | 4 | 292 | Arctic fox (Vulpes lagopus) | 3 |
| [43] | Pinigin, A. F. | 1969 | Russia | Asia | Serology | Farm | SAT | Brucella suis type 4 | 3 | 136 | Red fox (Vulpes vulpes) | 3 |
| [44] | Sachs, R | 1967 | Tanza-nia | Africa | Serology | Wild | SAT | Brucella abortus | 7 | 150 | Black-backed jackal (Canis mesomelas) | 2 |
| [44] | Sachs, R | 1967 | Tanza-nia | Africa | Serology | Wild | SAT | Brucella abortus | 10 | 30 | Wild dog (Lycaon pictus) | 2 |

**Note**: Reported prevalence of *Brucella* spp. in free-ranging and farmed wild canids, stratified by country and continent, with diagnostic category, test method, life condition, and host species. *Positive* indicates the number of test-positive animals, and *N (sample size)* indicates the total number of animals tested in each study arm. QA denotes the quality appraisal score based on an adapted seven-item Newcastle–Ottawa Scale (range 0–7), where higher values indicate lower risk of bias. Scores of 5–7 indicate low risk of bias, 3–4 moderate risk, and 0–2 high risk. **Abbreviations**: RBT = Rose Bengal Test; SAT = Standard Agglutination Test; BPAT = Buffered Plate Agglutination Test; CARD = Card Agglutination Test; RIV = Rapid Immunoversion Test; CFT = Complement Fixation Test; AGID = Agar Gel Immunodiffusion; CIE = Counterimmunoelectrophoresis; CIEF = Counterimmunoelectrophoresis (FPA-adapted variant); ICT = Immunochromatographic Test; FPA = Fluorescence Polarization Assay; ELISA = Enzyme-Linked Immunosorbent Assay; BBA = Buffered Brucella Antigen Agglutination; 2-ME = 2-Mercaptoethanol reduction test; IFA = indirect fluorescent antibody assay; Culture = bacterial culture; PCR = Polymerase Chain Reaction.

Study quality was not used as an inclusion criterion but was used to characterize the overall risk-of-bias profile of the evidence base and to contextualize regional and diagnostic-specific findings, and it was explicitly examined in sensitivity analyses. Detailed item-level scores for each NOS domain are provided in S2 Table.

### Outcomes and unit of analysis

Two outcomes were analyzed on separate diagnostic tracks:

1. **Exposure (serology):** misclassification-adjusted true seroprevalence.

2. **Active infection:** prevalence confirmed by PCR and/or culture (descriptive only, as confirmatory data were too limited for integration into the Bayesian modeling framework).

The unit of analysis was a "population" defined as the unique combination of Study ID (first author and publication year) × species × country × year × total sample size (N). Rows sharing identical parameters across all of these dimensions and N were treated as the same animals tested by different assays (multi-test populations). Rows differing in N, year, or species were treated as distinct populations. PCR/culture data were never combined with serology.

Analyses were performed for (i) wild-only datasets (primary) and (ii) wild plus farm/captive datasets (sensitivity analysis). "Farm/captive" status was based on the original study context.

### Serology track: misclassification-adjusted true seroprevalence

For each population $j$ tested with one or more serological assays $t$, we estimated a single true seroprevalence ($\theta_j$). Observed positives ($y_jt$) were modeled as:

$y_j t \sim Binomial(N_j, q_j t)$, where $q_j t = Se_t \cdot \theta_j + (1 - Sp_t)(1 - \theta_j)$.

where $Se_t$ and $Sp_t$ are assay-specific sensitivity and specificity [46]. When studies reported serial or parallel testing algorithms (for example "RBT screen followed by CFT confirm"), we used the implied combined sensitivity/specificity (serial: $Se = Se_1 Se_2$, $Sp = 1 - (1 - Sp_1)(1 - Sp_2)$; parallel: $Se = 1 - (1 - Se_1)(1 - Se_2)$, $Sp = Sp_1 Sp_2$). Otherwise, each assay contributed its own likelihood term.

True seroprevalence ($\theta_j$) was estimated using grid-based maximum likelihood estimation (MLE) with 95% profile-likelihood confidence intervals. Continent-level summaries were calculated as N-weighted means of $\theta_j$ with conservative weighted-variance approximations. Estimates are presented separately for wild-only and sensitivity datasets [47].

### Infection track: PCR/culture prevalence

Because confirmatory data were sparse, PCR and culture prevalence were summarized descriptively as pooled proportions (total positives/ total tested) within species, country, and continent strata, with 95% Wilson confidence intervals. Zero-event studies were retained.

### Machine learning Bayesian meta-analysis (serology)

To account for imperfect test performance and heterogeneity, we applied a hierarchical Bayesian measurement-error model. Observed positives per arm were modeled as binomial outcomes with apparent probabilities derived from assay Se/Sp applied to an underlying latent true prevalence. True prevalence was modeled on the logit scale with random effects for study and continent; species effects were added in stratified models.

When available, study-reported Se/Sp were used directly; otherwise, method-specific priors centered on literature-typical performance were applied. A beta–binomial extension addressed over-dispersion where necessary. Models were fitted using Hamiltonian Monte Carlo (No-U-Turn Sampler [NUTS]) with four chains, adaptive warm-up, and conservative

target acceptance. Convergence was verified via Gelman–Rubin statistics (R ≈ 1.00), effective sample sizes, and visual inspection of trace, rank, autocorrelation, and energy/ Bayesian fraction of missing information (BFMI) plots [47].

Model performance was assessed through posterior predictive checks and study-level calibration plots. Comparisons were made using Pareto-smoothed importance sampling leave-one-out cross-validation (PSIS-LOO) and widely applicable information criterion (WAIC); study influence was summarized using Pareto-k values. To enhance robustness, two safeguards were applied: (i) a Trim-K influence-weighted likelihood, and (ii) heavy-tailed priors on variance components. The combined robust+Trim-K model was reported alongside the baseline.

As a transparency check, we also generated Monte Carlo Rogan–Gladen estimates of true prevalence that propagate uncertainty in assay performance; these were consistent with Bayesian posteriors.

### Software and reproducibility

All analyses were scripted in Python and world heat map figure was constructed using R [48] Reproducible code and documentation are available via the public ShamsFarzane GitHub repository (https://github.com/ShamsFarzane/brucellosis-canids-seroprevalence.git).

## Results

### Search results

The database search yielded 4,120 records, including PubMed (n = 1,020), Scopus (n = 985), Web of Science (n = 910), Google Scholar (n = 720), and SciELO (n = 485). An additional 87 records were identified through reference list screening. After removal of 307 duplicates, 3,900 records remained for title and abstract screening. Of these, 3,810 were excluded because they did not report primary prevalence data, focused on domestic or captive animals, or involved non-canid host species. Ninety full-text articles were assessed for eligibility. Fifty-three articles were excluded for lack of original prevalence data (n = 18), focus on domestic or zoo animals (n = 13), unclear or invalid diagnostic methods (n = 9), unrepresentative or insufficient sample sizes (n = 8), or incomplete methodological reporting (n = 5). A detailed exclusion log is provided in S1 Table. A total of 37 studies met the inclusion criteria and were incorporated into the scoping review and quantitative synthesis. The study selection process is summarized in Fig 1. Inter-rater agreement during the calibration exercise was high (κ = 0.86).

### Characteristics of included studies

The evidence base comprised 37 studies conducted between 1962 and 2025, representing 48 wild canid populations from 16 countries across five continents. Geographic representation was uneven, with most data originating from North America, South America, and Europe, and more limited sampling in Asia and Africa. Sample sizes ranged from 10 to 997 individuals. Twelve wild canid species were represented, with the most frequently sampled taxa being *Canis lupus*, *Vulpes vulpes*, *Canis latrans*, *Cerdocyon thous*, *Vulpes lagopus*, and *Lycalopex culpaeus*.

Serological diagnostics included the Rose Bengal Test (RBT), Standard Agglutination Test (SAT), Buffered Plate Agglutination Test (BPAT), Card Agglutination Test (CARD), Complement Fixation Test (CFT), enzyme-linked immunosorbent assay (ELISA), agar gel immunodiffusion (AGID), counterimmunoelectrophoresis (CIE), fluorescence polarization assay (FPA), and immunochromatographic test (ICT). PCR and culture data were available from a smaller subset of studies. Study quality, assessed using the adapted Newcastle–Ottawa Scale, ranged from 1 to 7.

### Risk of bias assessment

Using predefined thresholds based on total Newcastle–Ottawa Scale scores, 21 studies were classified as low risk of bias, 12 as moderate risk, and 4 as high risk. Risk-of-bias patterns varied across regions, diagnostic approaches, and reporting

 

practices. Studies classified as moderate or high risk most frequently lacked explicit inclusion or exclusion criteria, sample-size justification, or detailed reporting of diagnostic cut-off values. High-risk studies commonly relied exclusively on older agglutination-based assays without confirmatory testing.

Geographically, lower-risk studies were more common in South America and North America, whereas studies from Africa and parts of Asia more often exhibited methodological limitations. Diagnostic modality was also associated with study quality, with investigations using ELISA, PCR, or CFT generally receiving higher appraisal scores than studies based solely on legacy agglutination tests. Quality scores for individual studies are reported in Tables 1 and S2. Full item-level scoring and a graphical summary of risk-of-bias domains are provided in S1 Fig.

**Global exposure is widespread, whereas confirmed active infection is uncommon**

Using a misclassification-aware model that integrates all serological assays performed on the same animals, we estimated the global true seroprevalence ($\theta$) of *Brucella* exposure in wild canids. Across 48 wild populations (N = 3,925 animals), the N-weighted global true seroprevalence was 8.2% (95% CI: 5.1–11.3%). Estimates varied widely across populations and continents, reflecting substantial heterogeneity in diagnostic methods, sampling frames, and time periods (S3 Table).

Confirmed active infection was rare. Pooled PCR and culture data across studies yielded 3.9% (95% Wilson CI: 3.0–5.1%; 52/1,325 positives). Because infection data were sparse and unevenly distributed (e.g., culture positives from Russia for *B. suis* biovar 4), these results are reported separately from serology.

When captive or farmed canids were included as a sensitivity analysis, the overall misclassification-adjusted seroprevalence increased to 15.5% (95% CI: 9.4–21.6%; 51 populations; N = 5,079), with high prevalence observed in blue-fox breeding farms. The infection proportion remained 3.9% since confirmatory tests were limited to wild settings.

Hierarchical Bayesian models correcting for imperfect diagnostic performance produced higher estimates. The baseline model estimated 23.1% (95% CrI: 19.4–27.1%); with influence trimming (Trim-K = 5), 15.8% (12.2–19.5%); with robust priors, 27.3% (22.6–32.5%); and with the combined robust+Trim-K specification, 21.7% (17.5–27.2%). Across model specifications, the estimated global true seroprevalence range was 16–27%, centered around 21–23%.

Posterior predictive checks and calibration plots demonstrated excellent model fit and convergence (no divergences; R ≈ 1.00; large effective sample sizes). Influence diagnostics confirmed that no single study dominated inference (Fig 2).

**Asia: sparse wild data, localized infection**

After excluding captive data, Asian wild canid data were extremely limited. Only one wild population (N = 56 golden jackals, Israel) was identified with 0/56 seropositive results. The corresponding misclassification-adjusted estimate was near zero, with a binomial 95% upper bound of ≈5.2%.

In contrast, culture-confirmed infection was documented in Russia, with pooled data showing 48/1,031 = 4.66% (95% Wilson CI: 3.53–6.12%) across wolves and Arctic foxes. Including captive populations (notably blue-fox farms in China) increased seroprevalence substantially but did not represent free-ranging wildlife.

The Bayesian model estimated a true seroprevalence of 4.6% (95% CrI: 4.0–5.5%). Diagnostic plots (energy/BFMI ≈ 0.92–0.96) and posterior predictive checks indicated good convergence. Estimates are shown in Fig 3.

**Europe: lowest exposure and limited infection evidence**

Europe showed the lowest exposure levels. Using the misclassification-aware model, the N-weighted true seroprevalence was ~0.8% (95% CI: 0.0–4.5%; 4 populations; N = 437).

PCR/culture data were scarce but present, yielding 4/254 = 1.6% (95% Wilson CI: 0.61–3.98%). The Bayesian model estimated a true seroprevalence of 1.6% (95% CrI: 1.3–1.9%) with strong convergence (R ≈ 1.00; energy/

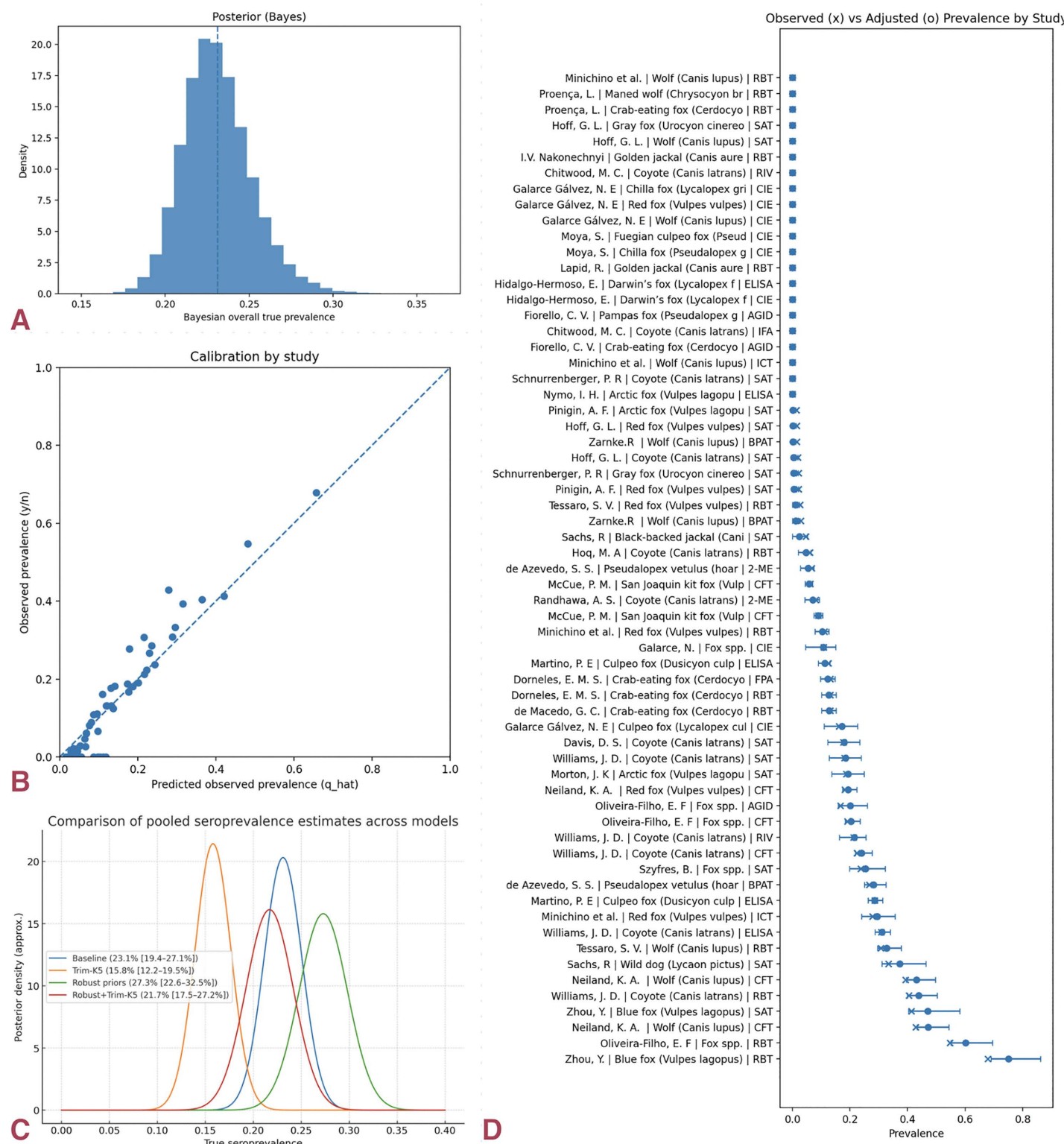

**Fig 2. Global true seroprevalence and model calibration.** (a) Posterior distribution of the misclassification-adjusted hierarchical Bayesian model for global *Brucella* seroprevalence in wild canids; the vertical line marks the posterior median. **(b)** Study-level calibration (observed vs. predicted) aligns with the 1:1 line, indicating good model fit. Chains converged (R ≈ 1.00; no divergences), and posterior predictive checks reproduced the empirical

distribution. **(c)** Sensitivity of pooled true seroprevalence to modeling choices (baseline, Trim-K, robust priors, robust+Trim-K). Influence trimming lowers estimates, while robust priors raise them; the combined model yields a stable mid-range (~22%). **(d)** Forest plot showing observed (×) versus misclassification-adjusted (●) prevalence with 95% confidence intervals (CIs) by diagnostic test. Marked heterogeneity is evident, with older, lower-specificity assays producing higher apparent prevalence, while PCR/culture arms remain low.

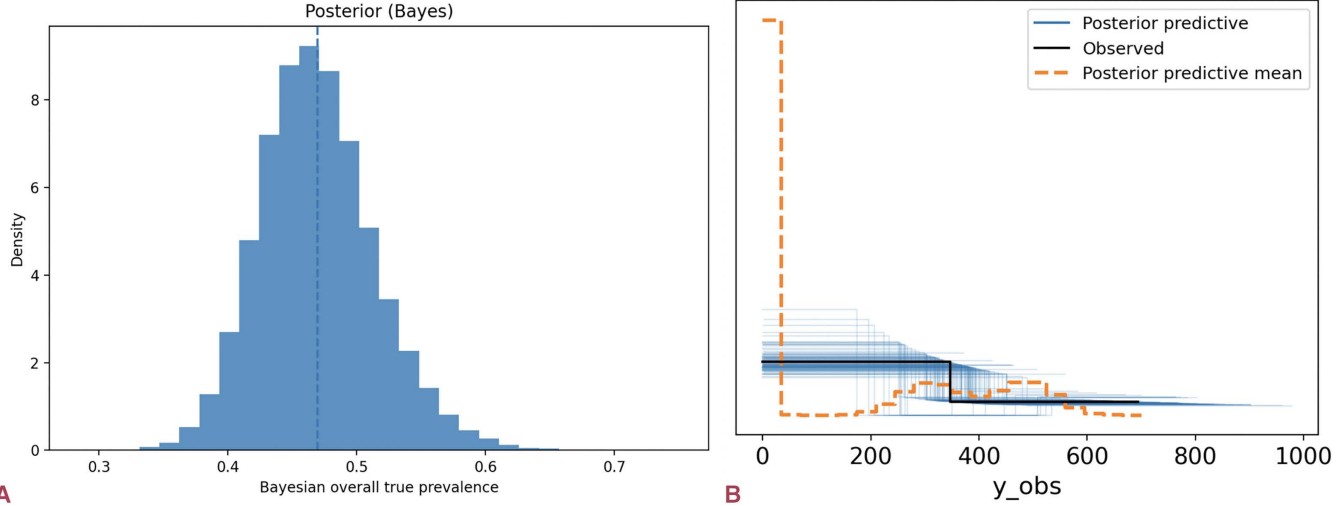

**Fig 3. True seroprevalence of brucellosis in Asian wild canids.** (a) Posterior distribution from the misclassification-adjusted hierarchical Bayesian model; the dashed line marks the posterior median (≈4.6%, 95% CrI 4.0–5.5%). (b) Posterior predictive check comparing replicated (blue) and observed (black) prevalence distributions. The model reproduces the central mass of the empirical data but slightly under-captures the tails, indicating an overall adequate though imperfect fit. Sparse data—driven mainly by one golden-jackal series—limit precision and highlight the need for additional wild sampling in Asia.

BFMI ≈ 0.86–0.92) and accurate posterior predictive fit (Fig 4). Positive detections were reported among wolves, Arctic foxes, and jackals, and were also observed in a subset of red fox studies.

### North America: low-to-moderate exposure, scarce confirmation

Across 22 North American wild populations (N = 2,066), the misclassification-aware model estimated ~4.1% true seroprevalence (95% CI: 0.5–7.8%). Exposure levels varied among species but remained generally low to moderate.

No PCR or culture-confirmed infections were found in the wild-only dataset. Bayesian models yielded pooled true seroprevalence estimates around 10%, depending on model specification, with satisfactory calibration and posterior predictive performance (Fig 5). These estimates indicate modest levels of serological exposure in North American wild canids.

### South America: higher exposure, limited confirmatory data

South America exhibited the highest apparent exposure. The misclassification-aware model estimated ~18% true seroprevalence (95% CI: 15–22%), consistently higher than in other continents. Confirmatory data were scarce: pooled infection results showed 0/40 PCR/culture positives (95% upper bound ≈7%).

The Bayesian baseline estimated true seroprevalence at 20.0% (95% CrI: 13.5–27.0%), and robust variants stabilized near 21.5% (18.0–26.5%). Diagnostic and influence analyses showed stable results without domination by individual studies (Fig 6). These results indicate higher serological prevalence estimates in South America relative to other regions, alongside limited confirmatory testing data.

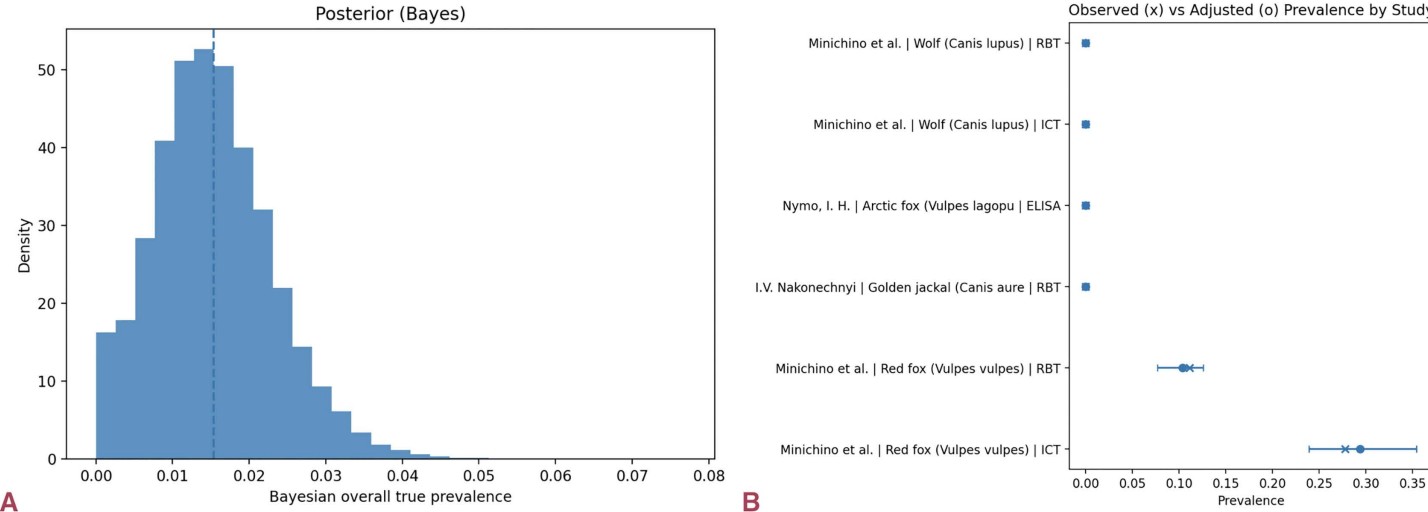

**Fig 4. Very low pooled prevalence with localized outliers in Europe.** (a) Posterior of true seroprevalence in Europe, with a median of ≈1.6% (95% CrI 1.3–1.9%). **(b)** Observed (×) versus misclassification-adjusted (●) prevalence by study. Most species are near zero; a few red-fox datasets drive the right-tail, reflecting local outliers rather than widespread exposure.

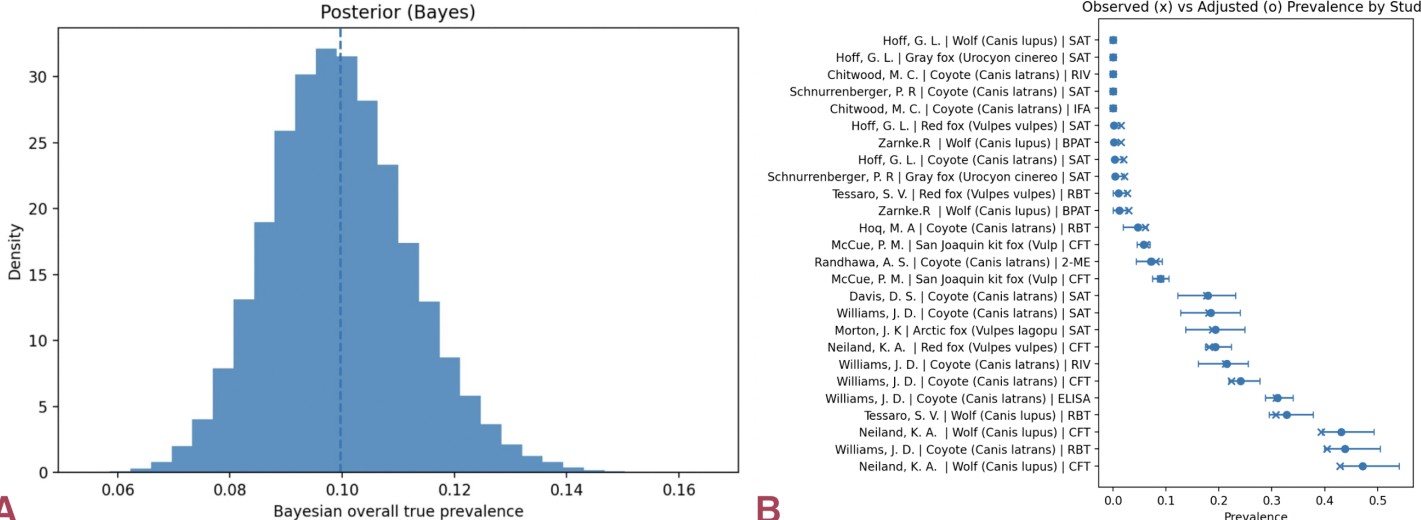

**Fig 5. Pooled true seroprevalence and study-level estimates in North America.** (a) Posterior distribution from the misclassification-adjusted hierarchical model; median true seroprevalence ≈10.0% (95% CrI 7.0–13.0%). The unimodal and moderately narrow posterior indicates intermediate exposure relative to other continents and good model pooling across studies. **(b)** Study-level apparent (×) and adjusted (●) prevalence with 95% CIs, labeled by diagnostic test. Coyote and red-fox series contribute most of the signal; older SAT/RBT assays tend to report higher apparent values, while adjusted estimates cluster lower, reflecting diagnostic-driven heterogeneity.

## Africa: under-sampled and uncertain

African data comprised only two populations from Tanzania (black-backed jackal and African wild dog). The estimated N-weighted true seroprevalence was ~6.0% (95% CI: 0.0–24.6%; k = 2; N = 180). The wide confidence interval reflects the small number of populations and limited sample size. All serological data were derived from SAT testing, and no PCR or

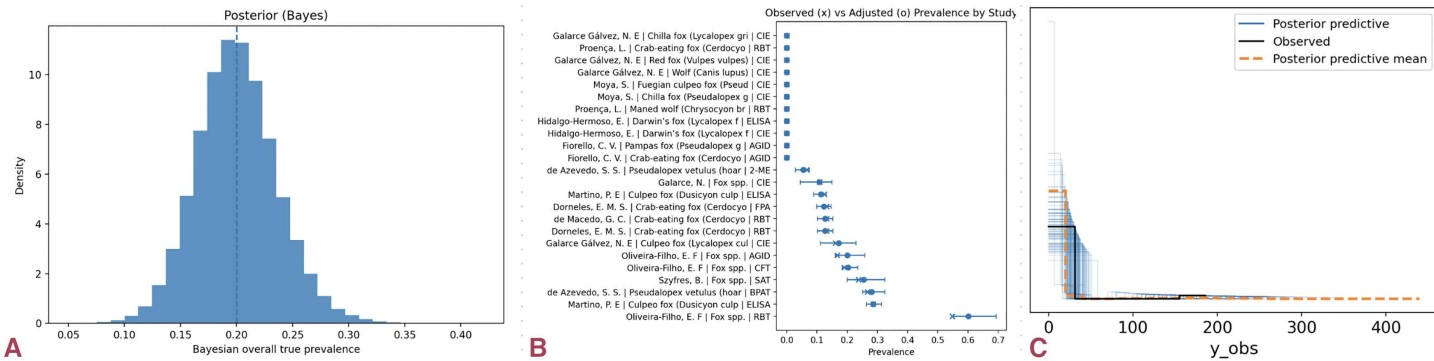

**Fig 6. Elevated true seroprevalence and diagnostic variability in South America.** (a) Posterior distribution from the misclassification-adjusted hierarchical model; median ≈21.5% (95% CrI ~ 18–26.5%). The unimodal, moderately tight posterior indicates consistently higher exposure than in North America or Europe. **(b)** Study-level apparent (×) and adjusted (●) prevalence with 95% CIs, labeled by diagnostic test. Crab-eating fox and culpeo series dominate the signal; SAT/RBT assays tend to overestimate compared to confirmatory tests. **(c)** Posterior predictive check for observed positives across study arms. Simulated (blue) distributions closely envelop empirical counts (black), and the posterior predictive mean (dashed) tracks the data, indicating good model fit.

culture-confirmed infections were reported for Africa. Despite adjustment for diagnostic performance, uncertainty remained high due to sparse data.

## Species patterns and diagnostic effects

Among species, coyotes (*Canis latrans*) showed the highest weighted seroprevalence (≈15.6%; 85/545).Higher seroprevalence estimates were observed for coyotes compared with other canid species.

Diagnostic usage strongly shaped observed patterns. SAT was the most common assay (~2,510 animals tested across 15 arms), while ELISA was used less frequently (~630 tested, 5 arms). Screening tests (RBT, SAT) consistently produced higher positivity than confirmatory tests (CFT, ELISA) across matched populations (Fig 7A).

Geographically, apparent seroprevalence in Asia was disproportionately influenced by data from farmed blue-fox populations in China (Fig 7C), reflecting captive rather than free-ranging systems, alongside concentrated sampling effort (Fig 7B). Confirmed infection remained limited and geographically restricted (Fig 7D).

## Discussion

Wild canids exhibit widespread serological exposure to *Brucella* spp. but active infection appears to be underestimated [3,46]. In our analysis among wild canids, integrating all serological assays and correcting for imperfect test performance, the estimated global true seroprevalence was 8.2% (95% CI: 5.1–11.3%). In contrast, only a small number of cases were confirmed by PCR or culture (3.9%; 95% CI: 3.0–5.1%), probably reflecting both the rarity of true infection and differences in testing availability between regions. Including farmed or captive populations raised seroprevalence to 15.5% (9.4–21.6%), largely due to outbreaks in blue-fox breeding facilities, demonstrating that captive amplification can distort regional patterns if not analyzed separately. Collectively, these findings indicate that *Brucella* exposure in wild canids is globally distributed but primarily localized around wildlife–livestock interfaces. Importantly, our use of a misclassification-aware, multi-assay modeling framework yields calibrated, decision-relevant prevalence estimates that reconcile heterogeneous diagnostic data, thereby addressing a persistent limitation in prior wildlife disease studies.

Seroprevalence differed across continents in consistent and biologically plausible ways. Europe showed the lowest exposure (~0.8%), with most wolves, jackals, and arctic foxes seronegative, while older red-fox data accounted for isolated outliers [3]. North America exhibited low-to-moderate exposure (~4.1%), consistent with localized transmission at

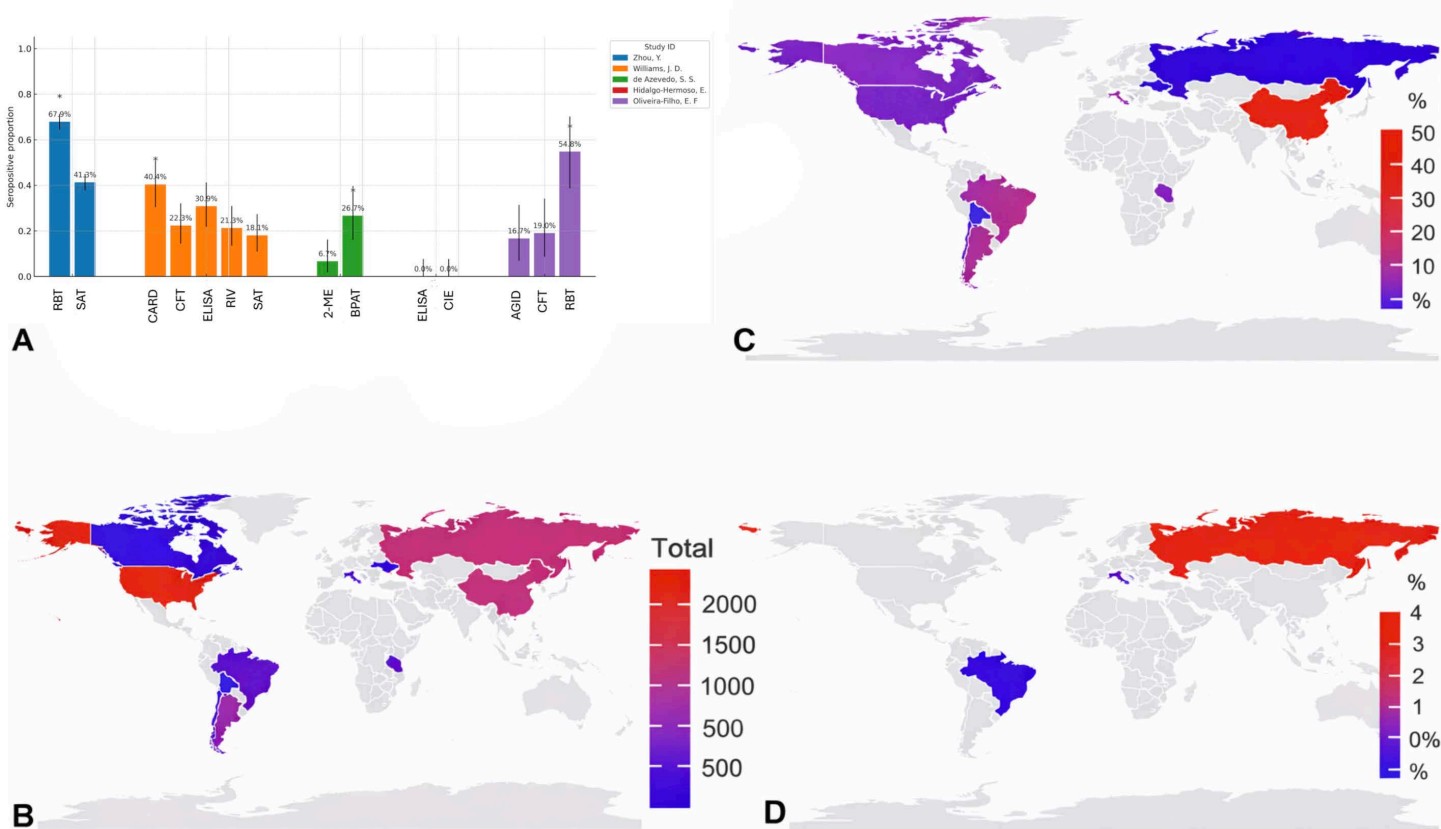

**Fig 7. Geographic clustering of seropositivity and assay-driven differences.** (a) For identical populations, bars show apparent seroprevalence (95% CIs); asterisks (*) denote significant differences. Screening assays (RBT, CARD, BPAT) consistently yield higher positivity than confirmatory tests (CFT, ELISA), underscoring diagnostic misclassification and the need for confirmatory validation. **(B)** Choropleth map of the total number of animals tested, aggregated by country across included studies (blue = lower totals; red = higher totals; grey = no data). **(C)** Choropleth map of apparent seroprevalence by country from serological studies (seropositive/total tested; blue = lower; red = higher; grey = no data). **(D)** Choropleth map of confirmed infection by country based on PCR and/or culture (blue = lower; red = higher; grey = no data). Maps were created in R using public-domain basemap shapefiles from Natural Earth (https://www.naturalearthdata.com).

livestock interfaces rather than broad enzootic circulation [3]. South America had higher values (~18%), largely in fox species from agro-pastoral landscapes, though confirmatory data (0/40 PCR or culture) remain absent [3]. Africa was severely under-sampled (~6%, N = 180; wide CI), limiting precision, [3] and Asia, when farm data were excluded, showed sparse wild serology (0/56 in jackals) but detectable infection in Russian culture data (4.66%) among wolves and arctic foxes, highlighting the danger of extrapolating from captive outbreaks [3,41]. Species composition also contributed: synanthropic canids, particularly coyotes (*Canis latrans*), showed the highest weighted seroprevalence (~15.6%), likely due to frequent contact with livestock environments, feeding on infected tissues, and shared resource use in peri-urban or agricultural landscapes [3,6]. Together, these patterns suggest that *Brucella* exposure in wild canids is regionally heterogeneous and often reflects ecological proximity to livestock rather than long-term maintenance within wild populations.

These patterns should be interpreted in light of the risk-of-bias assessment, which indicated greater methodological limitations in under-sampled regions and in studies relying exclusively on legacy serological assays. Much of the observed heterogeneity arises from methodological rather than biological causes. Agglutination-based screening assays such as RBT and SAT, consistently overestimate seropositivity relative to confirmatory tests such as CFT or ELISA [5,46].

Within-study comparisons revealed diagnostic inflation, with higher positivity reported for RBT than for CFT or AGID, and for CARD than for SAT or Rapid Immunoversion Test (RIV). After correcting misclassification and combining data from multiple assays, extreme values were reduced and regional estimates became more consistent. This suggests that apparent differences in naïve pooled results are driven mainly by variation in assay methods and cut-off thresholds, rather than by pathogen ecology alone [5]. The hierarchical Bayesian models,which explicitly correct for imperfect sensitivity and specificity and share information across studies, produced higher global seroprevalence estimates (approximately 16–27%,centered around 21–23%), as expected under diagnostic uncertainty and high heterogeneity. Posterior predictive checks, convergence diagnostics, and influence assessments all supported model stability, and robust or influence-trimmed variants yielded consistent results. The overall agreement between these Bayesian results and the misclassification-aware frequentist estimates suggests that much of the observed variation in seroprevalence is driven by differences in diagnostic methods rather than true ecological variation in *Brucella* circulation, resulting in more calibrated and decision-relevant estimates of global exposure in wild canids [3,46].

For One Health surveillance, programs should prioritize targeted sampling at livestock–wildlife interfaces, especially foxes and coyotes in agro-pastoral landscapes, and combine screening with confirmatory testing while explicitly modeling diagnostic misclassification to avoid bias [4,5]. Regions with elevated serology but little confirmatory data, including South America and parts of Asia, warrant targeted PCR and culture investigations, and captive systems, including fox farms in China, require strengthened biosecurity to prevent amplification and spillover [40]. By quantifying diagnostic uncertainty and translating results into a surveillance design framework, this study provides a reproducible model for decision-grade wildlife disease monitoring applicable across taxa and regions.

The main limitations of this study are uneven geographic and species coverage, particularly in Africa and parts of Asia; reliance in many studies on legacy agglutination assays with variable thresholds; and the scarcity of paired serology and PCR data from the same individuals. Standardized metadata collection, including age, carcass condition, vaccination history, and habitat context, alongside harmonized confirmatory testing protocols, would improve bias adjustment and strengthen inference on transmission potential. Future research should extend this framework to other wildlife reservoirs and zoonotic pathogens, integrating serological, molecular, and ecological data under unified Bayesian evidence synthesis. Such approaches will be critical for identifying true maintenance hosts and informing effective One Health interventions.

## Conclusion

Wild canids show widespread exposure to *Brucella* spp. while confirmed active infection is reported much less often, a pattern shaped not only by biology but also by important differences in diagnostic methods and geographic coverage. Our risk-of-bias assessment indicates that much of the apparent variation among regions and species reflects these methodological differences particularly the reliance on older assays and limited data from Africa and parts of Asia, rather than true ecological contrasts alone. After accounting for diagnostic error and study-level uncertainty through misclassification-aware hierarchical modeling, exposure appears globally distributed but uneven, with higher levels concentrated near wildlife–livestock interfaces and in intensively monitored settings. Overall, the evidence is most consistent with wild canids acting primarily as sentinels of environmental and livestock-associated *Brucella* exposure, while still allowing for localized maintenance in specific ecological or management contexts, especially in captive or high-contact systems. By integrating diverse diagnostic data and explicitly addressing bias, this study provides more reliable, decision-relevant estimates of global exposure and a transferable framework for improving wildlife disease surveillance and One Health monitoring.

### AI assistance disclosure

The authors used ChatGPT-5 (OpenAI) solely to assist with grammar, sentence structure, and clarity during manuscript revision. All scientific content, data analysis, interpretation, and conclusions were generated entirely by the authors. The authors reviewed and edited all AI-suggested language and take full responsibility for the final text.

## Supporting information

**S1 File. PRISMA-ScR Checklist.** Attribution: This PRISMA-ScR checklist is reproduced from the PRISMA-ScR statement materials and is licensed under the Creative Commons Attribution 4.0 International License (CC BY 4.0). The original guideline and supporting materials are available at https://www.prisma-statement.org/. The checklist is cited as: Tricco AC, Lillie E, Zarin W, O'Brien KK, Colquhoun H, Levac D, et al. PRISMA Extension for Scoping Reviews (PRISMA-ScR): Checklist and Explanation. Ann Intern Med. 2018;169(7):467–473. https://doi.org/10.7326/M18-0850.
(DOCX)

**S1 Fig. Risk-of-bias heat-map figure.**
(DOCX)

**S2 File. Full database search strings used across all bibliographic sources.**
(DOCX)

**S1 Table. Full-text exclusion log.**
(DOCX)

**S2 Table. Newcastle–Ottawa Quality Appraisal Scores for included studies.**
(DOCX)

**S3 Table. Study-level meta-analysis data for *Brucella* exposure and infection in wild canids.**
(DOCX)

## Author contributions

**Conceptualization:** Nahal Sarvestani, Armin Mirshahi, Mobina Pato, Arman Abdous.

**Data curation:** Aria Javani Farbod, Armina Khayatderafshi.

**Formal analysis:** Farzane Shams.

**Investigation:** Armin Mirshahi, Mobina Pato, Mobina Payami.

**Methodology:** Farzane Shams, Aria Javani Farbod, Armina Khayatderafshi, Arman Abdous.

**Project administration:** Arman Abdous.

**Software:** Farzane Shams.

**Supervision:** Farzane Shams, Mobina Pato, Arman Abdous.

**Visualization:** Aria Javani Farbod, Armina Khayatderafshi.

**Writing – original draft:** Farzane Shams, Mobina Pato, Arman Abdous.

**Writing – review & editing:** Nahal Sarvestani, Farzane Shams, Armin Mirshahi, Mobina Pato, Mobina Payami, Arman Abdous.

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
