## [Decision Letter · Decision Letter 0]

23 Nov 2025

From tests to truth: A misclassification-aware machine learning framework for estimating brucellosis seroprevalence in wild canids

Dear Dr. Abdous,

Thank you for submitting your manuscript to PLOS Neglected Tropical Diseases. After careful consideration, we feel that it has merit but does not fully meet PLOS Neglected Tropical Diseases's publication criteria as it currently stands. Therefore, we invite you to submit a revised version of the manuscript that addresses the points raised during the review process.

Please submit your revised manuscript within 3 months. If you will need more time than this to complete your revisions, please reply to this message or contact the journal office at plosntds@plos.org. Please include the following items when submitting your revised manuscript:

We look forward to receiving your revised manuscript.

Kind regards,

Richard A. Bowen, DVM PhD

Academic Editor

Ana LTO Nascimento

Section Editor

Shaden Kamhawi

co-Editor-in-Chief

Paul Brindley

co-Editor-in-Chief

**Additional Editor Comments:**

Your manuscript has been reviewed by 3 experts and each has provided suggestions and comments to improve your manuscript. Please evaluate their comments carefully, edit your manuscript to address those issues and resubmit with a "Responses to reviewers" document describing how you modified your manuscript and whether or not you agree with reviewer comments. Thank you.

**Journal Requirements:**

At this stage, the following Authors/Authors require contributions: Nahal Sarvestani, Armin Mirshahi, Mobina Pato, Aria Javani Farbod, Armina Khayatderafshi, Farzane Shams, Mobina Payami, and Arman Abdous. Please ensure that the full contributions of each author are acknowledged in the "Add/Edit/Remove Authors" section of our submission form.

Potential Copyright Issues:

i) Figure 6B. Please (a) provide a direct link to the base layer of the map (i.e., the country or region border shape) and ensure this is also included in the figure legend; and (b) provide a link to the terms of use / license information for the base layer image or shapefile. We cannot publish proprietary or copyrighted maps (e.g. Google Maps, Mapquest) and the terms of use for your map base layer must be compatible with our CC BY 4.0 license.

5) Please note that your Data Availability Statement is currently missing the DOI/accession number of each dataset OR a direct link to access each dataset. If your manuscript is accepted for publication, you will be asked to provide these details on a very short timeline. We therefore suggest that you provide this information now, though we will not hold up the peer review process if you are unable.

**Reviewers' Comments:**

Reviewer's Responses to Questions

**Key Review Criteria Required for Acceptance?**

**Methods**

-Are the objectives of the study clearly articulated with a clear testable hypothesis stated?

-Is the study design appropriate to address the stated objectives?

-Is the population clearly described and appropriate for the hypothesis being tested?

-Is the sample size sufficient to ensure adequate power to address the hypothesis being tested?

-Were correct statistical analysis used to support conclusions?

-Are there concerns about ethical or regulatory requirements being met?

Reviewer #1: yes

Reviewer #2: Hypothesis is clearly articulated, design appropriate, population size acceptable

Reviewer #3: 1. The authors state that they included studies that were "published between 1962 and 2025 in English, Persian, Spanish, Portuguese, or Russian." However, they selection of the period 1962 - 2025 is not justified. in the limitations, authors should acknowledge the language restrictions that could have limited the yield from literature search

2. In accordance with PRISMA-ScR guidelines, should provide the search terms and string for at least one data base. The current section on information sources lacks detail on how the records were searched, despite having referred to the supplementary material 2

3. Authors state that "Inter-rater agreement was assessed during a calibration exercise on a random subset of studies and was high. Reasons for full-text exclusion were documented and summarized in the PRISMA diagram." To enhance reporting transparency, authors should include another supplementary file that reasons for exclusion of each of the 53 studies at full text review. It's not clear how the inter-rater agreement was assessed and neither are there results in the results section.

4. In the data extraction section, authors state that "Records sharing identical Study ID" were excluded. It's not clear what identical means? This needs to be clarified on what features were used to qualify this. If Digital object identifiers were used, let it be stated for clarity

5. The description of how quality appraisal was done is limited and very unclear. How were the domains of the New-castle Ottawa Scale used to classify the overall quality as "Good", "Fair" or "poor"? In the associated supplementary file 3, what do the codes 1 and 0 mean? In the final "QA" column, what do those numerical values mean and what scaled was used to interpreted them. After doing the quality appraisal, how was risk of bias assessed, and how were the results of RoB analysis integrated in the final interpretation of results? Authors should provide a clear description in the methods. if possible, authors should consider adding a traffic light diagram as a supplementary file

**Results**

-Does the analysis presented match the analysis plan?

-Are the results clearly and completely presented?

-Are the figures (Tables, Images) of sufficient quality for clarity?

Reviewer #1: yes

Reviewer #2: Results are well presented

Reviewer #3: Before the synthesis of the results, authors should include the following sections to ensure complete reporting following the PRISMA-ScR guidelines

1. Search results (This section should show the numbers of sources of evidence screened, assessed for eligibility, and included in the review, with reasons for exclusions at each stage (refer to the supplementary material that shows reasons for exclusion of each of 53 studies excluded). Cite and include the PRISMA diagram with clear caption. In the description preceding the PRISMA diagram, cite all the studies included)

2. Characteristics of the studies included. (Authors should describe the studies included and probably refer to table 1 that needs to be improved. It's not clear what the column labelled "Total" means. This needs to be clarified. include the column of sample size. If the column labelled "QA" in table 1 means quality appraisal, in the table foot note, describe what those values mean)

3. Risk of Bias (Provide a synthesis of the findings from the risk of bias analysis. Clearly show which studies has low, moderate or high risk of bias). The integrate the results of the risk of bias analysis in the results discussion.

**Conclusions**

-Are the conclusions supported by the data presented?

-Are the limitations of analysis clearly described?

-Do the authors discuss how these data can be helpful to advance our understanding of the topic under study?

-Is public health relevance addressed?

Reviewer #1: no

Reviewer #2: Conclusions are supported by data and limitations addressed. It is clear how these data can be used to guide brucella surveillance, particularly in under-represented countries.

Reviewer #3: Revise the conclusions based on the results of risk of bias analysis

**Editorial and Data Presentation Modifications?**

Reviewer #1: (No Response)

Reviewer #2: There are problems with the citations.

1) The footnotes do not always follow the same convention with regard to punctuation. For example, on Page 5, three different footnote conventions. On line 90 "infection.[4]". On line 92 "settings. [5]". And on line 98 "hosts[3,6].". Please review all footnotes to ensure they all use the same convention. Most are like teh example from line 98.

2) There are several references that are incomplete.

For example, reference 30 is "Hoq MA. A serologic survey of Brucella agglutinins in wildlife and sheep. 1978." No journal, issue, volume or page number is given. An internet search identified this may have been published in California Veterinarian volume 32, pages 15-17.

Reference 35 is a doctoral thesis, and I am not sure the citation conforms to PLoS standards.

Reference 36, the author name is all caps, unlike other citations.

Reference 43 "Nymo IH, Fuglei E, Mørk T, Breines EM, Holmgren KE, Davidson RK, et al. Why are Svalbard Arctic foxes Brucella spp. seronegative? 2022." No journal, issue, volume or page number is given. An internet search identified this may have been published in Polar Research volume 41, page reference 7867.

References 47 and 48 also appear to be incomplete.

I did not exhaustively examine all references. As such, the authors should examine the entire References section and make sure all citations are complete and conform to PLoS standards.

Reviewer #3: In the abstract, line 50 - 51, clarify which exposure is being referred to. is it of humans or wild canids? In the methods section of the abstract, show how records were obtained and from which databases. By stating "Across 48 wild populations (N = 3,925 animals)", which ones are authors specifically referring to?

**Summary and General Comments**

Reviewer #1: This manuscript addresses an important and timely zoonotic disease question using an innovative misclassification-aware modeling framework, but several aspects of the methodology, data handling, interpretation, and presentation require clarification and strengthening to ensure robustness and transparency before the work is suitable for publication.

1.Please rewrite your abstcart into non structured abstract.

2.Please revise this sentence as it looks like AI generated “While most control and surveillance programs have historically focused on livestock and human infections, increasing evidence 78 indicates that wildlife—particularly free-ranging canids such as wolves (Canis lupus), foxes (Vulpes spp., Lycalopex spp.), jackals (Canis aureus), and coyotes (Canis latrans)—may play underappreciated roles in the ecology and transmission of Brucella species. ”.

3.Please delete all dots before all citations and relocate after the references throughout the entire manuscript.

4.Line 95; wrong reference citation style.

5.Although you have AI Disclosure, I suspect a lot of sentences are generated by AI, please revise the entire manuscript.

6.The introduction overstates the novelty of the modeling strategy without clearly distinguishing it from existing Bayesian disease-misclassification frameworks. Clarify how your approach differs from established latent-class or hierarchical misclassification models in wildlife epidemiology.

7.PCR and culture data are treated descriptively but not integrated into the modeling framework. Consider incorporating confirmatory data in a joint model or clearly justify why integration was not feasible.

8.The text asserts that canids do not act as maintenance hosts but this conclusion may exceed the strength of the available data. Rephrase conclusions to emphasize sentinel roles without excluding possible maintenance potential in specific contexts.

9.Several abbreviations (e.g., FPA, ICT, CIE) appear before being defined. Ensure all diagnostic abbreviations are defined at first mention.

10.The Results section includes interpretive statements better suited for the Discussion. Restrict results to data presentation and move interpretive narrative to the Discussion.

Reviewer #2: This is a nice, well-written manuscript that consolidates knowledge of Brucella distribution worldwide into an easy to reference format, and estimates frequencies of exposure/infection in canids. The implications for low coverage in specific regions and the association of canid brucellosis with nearby livestock are clear. The only concern is that the references are identified by inconsistent footnote punctuation and the citations are incomplete. This is easily correctable.

Reviewer #3: I would like to appreciate the authors for tackling an important and understudied topic on the seroprevalence of brucellosis in wild canids. They provide a rigorous global synthesis of Brucella exposure in wild canids, applying misclassification-aware machine learning to correct diagnostic bias. It offers calibrated seroprevalence estimates, highlights major geographic gaps, and delivers a reproducible framework that strengthens wildlife disease surveillance and One Health decision-making. However, the strength of the study needs to be improved by handling the reporting transparency needs highlighted in the review comments

PLOS authors have the option to publish the peer review history of their article (what does this mean? ). If published, this will include your full peer review and any attached files.

**Do you want your identity to be public for this peer review?** For information about this choice, including consent withdrawal, please see our Privacy Policy .

Reviewer #1: **Yes:** Ahmed Hamdy Ghonaim

Reviewer #2: **Yes:** H. Carl Gelhaus

Reviewer #3: No

**Figure resubmission:**
---

## [Decision Letter · Decision Letter 1]

11 Feb 2026

Dear Dr. Abdous,

We are pleased to inform you that your manuscript 'From tests to truth: A misclassification-aware machine learning framework for estimating brucellosis seroprevalence in wild canids' has been provisionally accepted for publication in PLOS Neglected Tropical Diseases.

Best regards,

Richard A. Bowen, DVM PhD

Academic Editor

Ana LTO Nascimento

Section Editor

Shaden Kamhawi

co-Editor-in-Chief

Paul Brindley

co-Editor-in-Chief

Thank you for the thoughtful revision of your manuscript. All reviewers now consider this a valuable contribution to the field and acceptable for publiction.

Reviewer's Responses to Questions

**Key Review Criteria Required for Acceptance?**

**Methods**

-Are the objectives of the study clearly articulated with a clear testable hypothesis stated?

-Is the study design appropriate to address the stated objectives?

-Is the population clearly described and appropriate for the hypothesis being tested?

-Is the sample size sufficient to ensure adequate power to address the hypothesis being tested?

-Were correct statistical analysis used to support conclusions?

-Are there concerns about ethical or regulatory requirements being met?

Reviewer #1: Yes

Reviewer #3: The revised manuscript now clearly follows the PRISMA-ScR guidelines, and the authors have demonstrated how they performed the risk of bias evaluation in the included studies and their final quality appraisal. The search string has been added and referenced

**Results**

-Does the analysis presented match the analysis plan?

-Are the results clearly and completely presented?

-Are the figures (Tables, Images) of sufficient quality for clarity?

Reviewer #1: Yes

Reviewer #3: Table 1 has been well clarified, plus all the results on quality of appraisal and risk of bias analysis.

**Conclusions**

-Are the conclusions supported by the data presented?

-Are the limitations of analysis clearly described?

-Do the authors discuss how these data can be helpful to advance our understanding of the topic under study?

-Is public health relevance addressed?

Reviewer #1: Yes

Reviewer #3: The authors conclusion on the exposure of wild canids to brucellosis is well supported by their data

**Editorial and Data Presentation Modifications?**

Reviewer #1: (No Response)

Reviewer #3: (No Response)

**Summary and General Comments**

Reviewer #1: The quality of the current manuscript has been improved greatly. I now agree for further process. Congratulations

Reviewer #3: I thank the authors for significantly improving this manuscript. Its now well aligned to the reporting standard they applied (PRISMA-ScR) and contributes to our further understanding of the role plaid by wild canids in the transmission of brucellosis

PLOS authors have the option to publish the peer review history of their article (what does this mean? ). If published, this will include your full peer review and any attached files.

**Do you want your identity to be public for this peer review?** For information about this choice, including consent withdrawal, please see our Privacy Policy .

Reviewer #1: **Yes:** Ahmed Hamdy Ghonaim

Reviewer #3: No

---

## [Editor Report · Acceptance letter]

Dear Dr Abdous,

We are delighted to inform you that your manuscript, "From tests to truth: A misclassification-aware machine learning framework for estimating brucellosis seroprevalence in wild canids," has been formally accepted for publication in PLOS Neglected Tropical Diseases.

Best regards,

Shaden Kamhawi

co-Editor-in-Chief

Paul Brindley

co-Editor-in-Chief
